# Exploiting Hankel–Toeplitz Structures
# for Fast Computation of Kernel Precision Matrices

**Frida Viset** †                                                     *f.m.viset@tudelft.nl*
*Delft Center for Systems and Control*
*Delft University of Technology, The Netherlands*

**Anton Kullberg** †                                                  *anton.kullberg@liu.se*
*Department of Electrical Engineering*
*Linköping University, Sweden*

**Frederiek Wesel**                                                   *f.wesel@tudelft.nl*
*Delft Center for Systems and Control*
*Delft University of Technology, The Netherlands*

**Arno Solin**                                                        *arno.solin@aalto.fi*
*Department of Computer Science*
*Aalto University, Finland*

**Reviewed on OpenReview:** *https://openreview.net/forum?id=s9ylaDLvdO*

## Abstract

The *Hilbert–space Gaussian Process* (HGP) approach offers a hyperparameter-independent basis function approximation for speeding up *Gaussian Process* (GP) inference by projecting the GP onto $M$ basis functions. These properties result in a favorable data-independent $\mathcal{O}(M^3)$ computational complexity during hyperparameter optimization but require a dominating one-time precomputation of the precision matrix costing $\mathcal{O}(NM^2)$ operations. In this paper, we lower this dominating computational complexity to $\mathcal{O}(NM)$ with *no additional approximations*. We can do this because we realize that the precision matrix can be split into a sum of Hankel–Toeplitz matrices, each having $\mathcal{O}(M)$ unique entries. Based on this realization we propose computing only these unique entries at $\mathcal{O}(NM)$ costs. Further, we develop two theorems that prescribe sufficient conditions for the complexity reduction to hold generally for a wide range of other approximate GP models, such as the *Variational Fourier Feature* (VFF) approach. The two theorems do this with no assumptions on the data and no additional approximations of the GP models themselves. Thus, our contribution provides a pure speed-up of several existing, widely used, GP approximations, *without further approximations*.

## 1 Introduction

*Gaussian Processes* (GPs, Rasmussen & Williams, 2006) provide a flexible formalism for modeling functions which naturally allows for the incorporation of prior knowledge and the production of uncertainty estimates in the form of a predictive distribution. Typically a GP is instantiated by specifying a prior mean and covariance (kernel) function, which allows for incorporation of prior knowledge. When data becomes available, the GP can then be conditioned on the observations, yielding a new GP which can be used for predictions and uncertainty quantification.

---

†-These authors contributed equally to this work. A reference implementation built on top of GPJax is available at https://github.com/AOKullberg/hgp-hankel-structure.

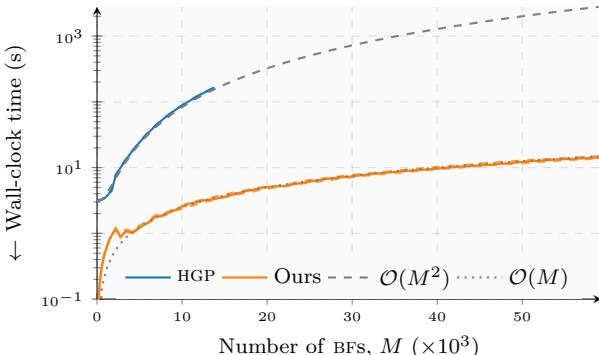

Figure 1: **An order of magnitude speed-up without any additional approximations:** Wall-clock time to compute the precision matrix for an increasing number $M$ of basis functions.

While all these operations have closed-form expressions for regression, the computations of the mean and covariance of the predictive GP require instantiating and inverting the kernel (Gram) matrix, which encodes pair-wise similarities between all data. These operations require respectively $\mathcal{O}(N^2)$ and $\mathcal{O}(N^3)$ computations, where $N$ is the number of data points. Furthermore, if one wishes to optimize the hyperparameters of the kernel function, which in GPs is typically accomplished by optimizing the log-marginal likelihood, the kernel matrix needs to be instantiated and inverted multiple times, further hindering the applicability of GPs to large-scale datasets.

The ubiquitous strategy to reduce this computational complexity consists in approximating the kernel matrix in terms of *Basis Functions* (BFs), yielding what is essentially a low-rank or "sparse" approximation of the kernel function (see Rasmussen & Williams, 2006; Quiñonero-Candela & Rasmussen, 2005; Snelson & Ghahramani, 2005). Computational savings can then be achieved by means of the matrix inversion lemma, which requires instantiating and inverting the *precision matrix* (sum of the outer products of the BFs) instead of the kernel matrix at prediction, thereby lowering the computational complexity of inference to $\mathcal{O}(NM^2 + M^3)$, where $M$ is the number of BFs. If the number of BFs is chosen smaller than the number of data points in the training set ($M < N$), computational benefits arise and the computational costs for hyperparameter optimization and inference are dominated by $\mathcal{O}(NM^2)$.

It is less widely known that this cost can be improved further. A notable BF framework is the HGP (Solin & Särkkä, 2020) which projects the GP onto a dense set of orthogonal, hyperparameter-independent BFs. This deterministic approximation is particularly attractive as it exhibits fast convergence guarantees to the full GP in terms of the number of BFs for smooth shift-invariant kernels compared to other approximations. Furthermore, the fact that the BFs are hyperparameter-independent speeds up GP hyperparameter optimization considerably, which in the HGP requires only $\mathcal{O}(M^3)$ operations after a one-time precomputation of the precision matrix, costing $\mathcal{O}(NM^2)$. These favorable properties have ensured a relatively wide usage of the HGP (see e.g. Svensson et al., 2016; Berntorp, 2021; Kok & Solin, 2018), and it is available in, e.g., PyMC (Abril-Pla et al., 2023) and Stan (Riutort-Mayol et al., 2022). However, as argued by Lindgren et al. (2022), a high number of BFs may be required for a faithful approximation of the full model. Thus, the HGP is typically employed in applications where forming the initial $\mathcal{O}(NM^2)$ projection does not become too heavy.

In this paper, we reduce this complexity to $\mathcal{O}(NM)$ with *no additional approximations* (see Fig. 1), by exploiting structural properties of the precision matrix. We remark that these structural properties arise from the BFs that are used to approximate the kernel, *not* the kernel itself. Our contributions are as follows.

- We show that the HGP precision matrix using basis functions defined on a (hyper-)cubical domain can be split in a sum of multilevel block-Hankel and a multilevel block-Toeplitz matrix (Fig. 2), where each summand only has $\mathcal{O}(M)$ unique entries instead of $\mathcal{O}(M^2)$. This allows us to determine the elements of the full precision matrix at $\mathcal{O}(NM)$ instead of $\mathcal{O}(NM^2)$.

- This method does not only hold for HGP basis functions. We provide sufficient conditions on the BFs for this reduction in computational complexity to hold. Examples of other BFs where the

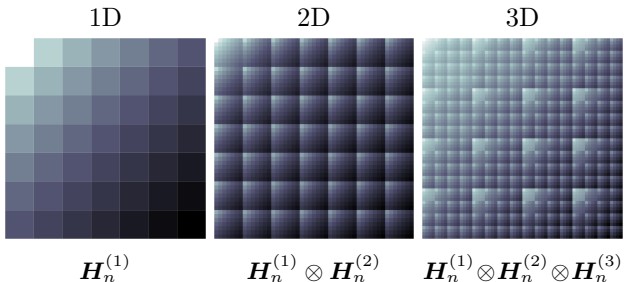

Figure 2: The precision matrix for polynomial basis functions has a nested Hankel structure. The visualization of the matrix is proportionally darker as the logarithm of each entry increases. The matrices are computed as the sum of all entries $\boldsymbol{H}_n$ for $n = \{1, \ldots, N\}$, where the expression for $\boldsymbol{H}_n$ is given below each matrix.

conditions hold, and that therefore can be sped up using the same technique are polynomial, (complex) exponential and (co)sinusoidal basis functions. We, therefore, enable the speed-up of other BF-based GP approximations such as variational Fourier features (Hensman et al., 2017).

In the experiments, we demonstrate in practice that our approach lowers the computational complexity of the HGP by an order of magnitude on simulated and real data.

## 2 Background

A GP is a collection of random variables, any finite number of which have a joint Gaussian distribution (Rasmussen & Williams, 2006). We denote a zero–mean GP by $f \sim \mathcal{GP}(0, \kappa(\cdot, \cdot))$, where $\kappa(\boldsymbol{x}, \boldsymbol{x}') : \mathbb{R}^D \times \mathbb{R}^D \to \mathbb{R}$ is the kernel, representing the covariance between inputs $\boldsymbol{x}$ and $\boldsymbol{x}'$. Given a dataset of input–output pairs $\{(\boldsymbol{x}_n, y_n)\}_{n=1}^N$, GPs are used for non-parametric regression and classification by coupling the latent functions $f$ with observations through a likelihood model $p(\boldsymbol{y} \mid f) = \prod_{i=1}^N p(y_i \mid f(\boldsymbol{x}_i))$.

For notational simplicity, we will in the following focus on GP models with a Gaussian (conjugate) likelihood, $y_i \sim \mathcal{N}(f(\boldsymbol{x}_i), \sigma^2)$. The posterior GP, $\mathcal{GP}(\mu_\star(\cdot), \Sigma_\star(\cdot, \cdot))$ can be written down in closed form by

$$\mu_\star(\boldsymbol{x}_\star) = \boldsymbol{k}_\star^\top (\boldsymbol{K} + \sigma^2 \boldsymbol{I})^{-1} \boldsymbol{y}, \tag{1a}$$

$$\Sigma_\star(\boldsymbol{x}_\star, \boldsymbol{x}'_\star) = \boldsymbol{k}(\boldsymbol{x}_\star, \boldsymbol{x}'_\star) - \boldsymbol{k}_\star^\top (\boldsymbol{K} + \sigma^2 \boldsymbol{I})^{-1} \boldsymbol{k}_{\star'}, \tag{1b}$$

where $\boldsymbol{K} \in \mathbb{R}^{N \times N}$ and $\boldsymbol{k}_\star \in \mathbb{R}^N$ are defined element-wise as $\boldsymbol{K}_{i,j} \coloneqq \kappa(\boldsymbol{x}_i, \boldsymbol{x}_j)$ and $[\boldsymbol{k}_\star]_i \coloneqq \kappa(\boldsymbol{x}_i, \boldsymbol{x}^\star)$ for $i, j \in 1, 2, \ldots, N$, and $\boldsymbol{k}_{\star'}$ is defined similarly. Observations are collected into $\boldsymbol{y} \in \mathbb{R}^N$. Due to the inverse of $(\boldsymbol{K} + \sigma^2 \boldsymbol{I})$ in the posterior mean and covariance, the computational cost of a standard GP scales as $\mathcal{O}(N^3)$, which hinders applicability to large datasets.

### 2.1 Basis Function Approximations

The prevailing approach in literature to circumvent the $\mathcal{O}(N^3)$ computational bottleneck is to approximate the GP with a sparse approximation, using a finite number of either inducing points or *Basis Functions* (BFs) (e.g., Rasmussen & Williams, 2006; Quiñonero-Candela & Rasmussen, 2005; Hensman et al., 2017). The BF representation is commonly motivated by the approximation

$$\kappa(\boldsymbol{x}, \boldsymbol{x}') \approx \boldsymbol{\phi}(\boldsymbol{x})^\top \boldsymbol{\Lambda} \, \boldsymbol{\phi}(\boldsymbol{x}'). \tag{2}$$

We use the notation from Solin & Särkkä (2020) to align with the next section. Here, $\boldsymbol{\phi}(\cdot) : \mathbb{R}^{\mathbb{D}} \to \mathbb{R}^M$ are the BFs, $\boldsymbol{\phi}(\cdot) \coloneqq [\phi_1(\cdot), \phi_2(\cdot), \ldots, \phi_M(\cdot)]^\top$. Further, $\boldsymbol{\Lambda} \in \mathbb{R}^{M \times M}$ are the corresponding BF weights. Combining this approximation with the posterior GP, Eq. (1), and applying the Woodbury matrix inversion lemma yields

$$\mu_\star(\boldsymbol{x}_\star) = \boldsymbol{\phi}(\boldsymbol{x}_\star)^\top \left( \boldsymbol{\Phi}^\top \boldsymbol{\Phi} + \sigma^2 \boldsymbol{\Lambda}^{-1} \right)^{-1} \boldsymbol{\Phi}^\top \boldsymbol{y}, \tag{3a}$$

$$\Sigma_\star(\boldsymbol{x}_\star, \boldsymbol{x}'_\star) = \sigma^2 \boldsymbol{\phi}(\boldsymbol{x}_\star)^\top \left( \boldsymbol{\Phi}^\top \boldsymbol{\Phi} + \sigma^2 \boldsymbol{\Lambda}^{-1} \right)^{-1} \boldsymbol{\phi}(\boldsymbol{x}'_\star). \tag{3b}$$

Here, $\boldsymbol{\Phi} \in \mathbb{R}^{N \times M}$ is commonly referred to as the regressor matrix and is defined as $\boldsymbol{\Phi}_{i,:} \coloneqq \boldsymbol{\phi}(\boldsymbol{x}_i)^\top$. Further, $\boldsymbol{\Phi}^\top \boldsymbol{\Phi} \in \mathbb{R}^{M \times M}$ is the *precision matrix* which is a central component of the following section. Computing this precision matrix requires $\mathcal{O}(NM^2)$ operations. With this approximation, computing the posterior mean and covariance in Eq. (3) requires instantiating and inverting $\left(\boldsymbol{\Phi}^\top \boldsymbol{\Phi} + \sigma^2 \boldsymbol{\Lambda}^{-1}\right)^{-1}$ which can be performed with $\mathcal{O}(NM^2 + M^3)$ operations. If we have more samples than BFs (which is the required condition for BFs to give computational savings), i.e., $N \geq M$, the overall computational complexity is then of $\mathcal{O}(NM^2)$, i.e., the inversion costs are negligible as compared to the cost of computing the precision matrix.

## 3 Methods

Our main findings are in the form of two theorems (Theorems 3.1 and 3.4). These theorems prescribe the necessary conditions that BF expansions need to fulfill to be able to reduce the computational complexity of computing the precision matrix $\boldsymbol{\Phi}^\top \boldsymbol{\Phi}$ from $\mathcal{O}(NM^2)$ to $\mathcal{O}(NM)$, applicable to multiple previous works that rely on parametric basis functions (incl. Lázaro-Gredilla et al., 2010; Hensman et al., 2017; Solin & Särkkä, 2020; Tompkins & Ramos, 2018; Dutordoir et al., 2020). Since our contribution reduces the cost of computing the precision matrix to $\mathcal{O}(NM)$, it follows that the computational complexity of computing the posterior mean or covariance is also of $\mathcal{O}(NM)$. Further, both of the theorems reduce the memory scaling from $\mathcal{O}(M^2)$ to $\mathcal{O}(M)$. Note that these reductions are *without* approximations, only relying on the structural properties of the considered models.

In the following, we assume that the kernel is a tensor product kernel (Rasmussen & Williams, 2006), i.e., $\kappa(x, x') = \prod_{d=1}^D \kappa^{(d)}(x^{(d)}, x^{(d)'})$, where $\kappa^{(d)}(\cdot, \cdot)$ is the kernel along the $d^{\text{th}}$ dimension. Then, if each component of the kernel is approximated with $m_d$ BFs such that

$$\kappa^{(d)}(x^{(d)}, x^{(d)'}) \approx \boldsymbol{\phi}^{(d)}(x^{(d)})^\top \boldsymbol{\Lambda}^{(d)} \boldsymbol{\phi}^{(d)}(x^{(d)'}), \tag{4}$$

where $\boldsymbol{\phi}^{(d)}(\cdot) : \mathbb{R} \to \mathbb{R}^{m_d}$ are the BFs $[\phi_1^{(d)}, \phi_2^{(d)}, \ldots, \phi_{m_d}^{(d)}]^\top$ along the $d^{\text{th}}$ dimension and $\boldsymbol{\Lambda}^{(d)} \in \mathbb{R}^{m_d \times m_d}$ contains the associated weights. The full kernel can then be approximated as

$$\kappa(\boldsymbol{x}, \boldsymbol{x}') \approx \prod_{d=1}^D \boldsymbol{\phi}^{(d)}(x^{(d)})^\top \boldsymbol{\Lambda}^{(d)} \boldsymbol{\phi}^{(d)}(x^{(d)'}), \tag{5}$$

where in this case $\boldsymbol{\phi}(\cdot) : \mathbb{R}^D \to \mathbb{R}^M$ and $\boldsymbol{\Lambda} \in \mathbb{R}^{M \times M}$ are

$$\boldsymbol{\phi}(\boldsymbol{x}) = \otimes_{d=1}^D \boldsymbol{\phi}^{(d)}(x^{(d)}), \tag{6a}$$

$$\boldsymbol{\Lambda} = \otimes_{d=1}^D \boldsymbol{\Lambda}^{(d)}. \tag{6b}$$

Here, $M \coloneqq \prod_{d=1}^D m_d$ is the total number of BFs. Given this decomposition, the precision matrix can be expressed as

$$\boldsymbol{\Phi}^\top \boldsymbol{\Phi} = \sum_{n=1}^N \boldsymbol{\phi}(\boldsymbol{x}_n) \boldsymbol{\phi}(\boldsymbol{x}_n)^\top = \sum_{n=1}^N \otimes_{d=1}^D \boldsymbol{\phi}^{(d)}(x_n^{(d)}) \left[\boldsymbol{\phi}^{(d)}(x_n^{(d)})\right]^\top. \tag{7}$$

This decomposition of the precision matrix is key in the following and we will primarily study the individual products $\boldsymbol{\phi}^{(d)}(x_n^{(d)})[\boldsymbol{\phi}^{(d)}(x_n^{(d)})]^\top$ where certain structure may appear that is exploitable to our benefit. To provide some intuition, we consider the precision matrix for polynomial BFs in 1D, 2D, and 3D (see Fig. 2). The 1D case (left) has a *Hankel* structure and the 2D (middle) and 3D (right) cases have 2-level and 3-level *block Hankel* structure, respectively. It is these types of structures that allow a reduction in complexity *without* approximations. Next, we provide clear technical definitions of the matrix structures and then proceed to state our main findings. All of the discussed matrix structures are also visually explained in Table A1 in App. E.

### 3.1 Hankel and Toeplitz Matrices

An $m \times m$ matrix $\boldsymbol{H}$ has Hankel structure iff it can be expressed using a vector $\boldsymbol{\gamma} \in \mathbb{R}^{(2m-1)}$ containing all the unique entries, according to

$$\boldsymbol{H} = \begin{bmatrix} \gamma_1 & \gamma_2 & \cdots & \gamma_m \\ \gamma_2 & \gamma_3 & \cdots & \gamma_{m+1} \\ \vdots & \vdots & \ddots & \vdots \\ \gamma_m & \gamma_{m+1} & \cdots & \gamma_{2m-1} \end{bmatrix}. \tag{8}$$

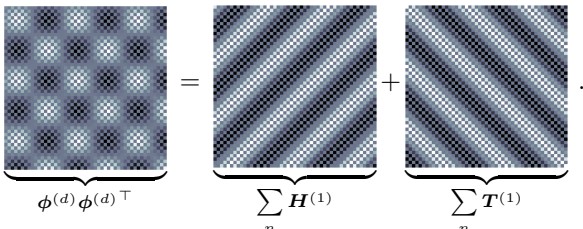

Figure 3: The precision matrix for sinusoidal basis functions in one dimension has neither Hankel nor Toeplitz structure. However, it can be decomposed into a sum of two matrices, where one has a Hankel structure, and one has Toeplitz structure. Here, 49 BFs are placed along one dimension.

In other words, each element of the Hankel matrix on the $i^{\text{th}}$ row and the $j^{\text{th}}$ column is given by $\boldsymbol{\gamma}_{i+j-1}$. Similarly, an $m \times m$ matrix has Toeplitz structure iff it can be expressed using a vector $\boldsymbol{\gamma} \in \mathbb{R}^{(2m-1)}$ such that each element on the $i^{\text{th}}$ row and the $j^{\text{th}}$ column is given by $\boldsymbol{\gamma}_{i-j+m}$. See Fig. 3 for an example of what a Hankel and a Toeplitz matrix visually looks like.

We define a matrix $\boldsymbol{H}^{(D)}$ as a $D$-level block Hankel matrix if it can be expressed as

$$\boldsymbol{H}^{(D)} = \begin{bmatrix} \boldsymbol{H}_1^{(D-1)} & \boldsymbol{H}_2^{(D-1)} & \dots & \boldsymbol{H}_{m_D}^{(D-1)} \\ \boldsymbol{H}_2^{(D-1)} & \boldsymbol{H}_3^{(D-1)} & \dots & \boldsymbol{H}_{m_D+1}^{(D-1)} \\ \vdots & \vdots & \ddots & \vdots \\ \boldsymbol{H}_{m_D}^{(D-1)} & \boldsymbol{H}_{m_D+1}^{(D-1)} & \dots & \boldsymbol{H}_{2m_D-1}^{(D-1)} \end{bmatrix}, \tag{9}$$

where $\boldsymbol{H}_k^{(D-1)}$ is a block Hankel matrix, and $\boldsymbol{H}_k^{(1)}$ is a simple Hankel matrix. Each submatrix $\boldsymbol{H}_k^{(d)}$ for $d \in \{1, \dots, D\}$ can be indexed by row and column $i_d, j_d \in \{1, \dots, m_d\}$ which again yields a submatrix $\boldsymbol{H}_{i_d+j_d-1}^{(d-1)}$ which can be indexed in the same manner. Therefore, an equivalent definition of a block Hankel matrix is that each individual entry of $\boldsymbol{H}^{(D)}$, given by a set of indices $i_1, j_1, \dots, i_D, j_D$, is given by the entry $\boldsymbol{\gamma}_{k_1,\dots,k_D}$ of a tensor $\boldsymbol{\gamma} \in \mathbb{R}^{K_1 \times K_2 \times \dots \times K_D}$, where $k_d = i_d + j_d - 1$, and $K_d = 2m_d - 1$. Similarly, $\boldsymbol{T}^{(D-1)}$ is a block Toeplitz matrix if each entry $i_1, j_1, \dots, i_D, j_D$ is given by the entry $\boldsymbol{\gamma}_{k_1,\dots,k_D}$ of a tensor $\boldsymbol{\gamma} \in \mathbb{R}^{K_1 \times K_2 \times \dots \times K_D}$, where $k_d = m_d + i_d - j_d$, and $K_d = 2m_d - 1$.

## 3.2 Block Hankel–Toeplitz Matrices

In addition to block Hankel and block Toeplitz structures, we require a slightly more general but highly related structure. We call this structure block Hankel–Toeplitz structure and define it as follows. A matrix $\boldsymbol{G}^{(D)}$ has a $D$-level block Hankel–Toeplitz structure, if the matrix is defined either as

$$\boldsymbol{G}^{(D)} = \begin{bmatrix} \boldsymbol{G}_1^{(D-1)} & \boldsymbol{G}_2^{(D-1)} & \dots & \boldsymbol{G}_{m_D}^{(D-1)} \\ \boldsymbol{G}_2^{(D-1)} & \boldsymbol{G}_3^{(D-1)} & \dots & \boldsymbol{G}_{m_D+1}^{(D-1)} \\ \vdots & \vdots & \ddots & \vdots \\ \boldsymbol{G}_{m_D}^{(D-1)} & \boldsymbol{G}_{m_D+1}^{(D-1)} & \dots & \boldsymbol{G}_{2m_D-1}^{(D-1)} \end{bmatrix}, \tag{10}$$

if level $D$ is Hankel, or as

$$\boldsymbol{G}^{(D)} = \begin{bmatrix} \boldsymbol{G}_{m_D}^{(D-1)} & \dots & \boldsymbol{G}_2^{(D-1)} & \boldsymbol{G}_1^{(D-1)} \\ \boldsymbol{G}_{m_D+1}^{(D-1)} & \dots & \boldsymbol{G}_3^{(D-1)} & \boldsymbol{G}_2^{(D-1)} \\ \vdots & \ddots & \vdots & \vdots \\ \boldsymbol{G}_{2m_D-1}^{(D-1)} & \dots & \boldsymbol{G}_{m_D+1}^{(D-1)} & \boldsymbol{G}_{m_D}^{(D-1)} \end{bmatrix}, \tag{11}$$

if level $D$ is Toeplitz. Further, $\boldsymbol{G}_j^{(D-1)}$ are $D-1$ level block Hankel–Toeplitz matrices if $D-1 > 2$, and a simple Hankel or Toeplitz matrix if $D-1 = 1$. Each submatrix $\boldsymbol{G}_k^{(d)}$ for $d \in \{1, \dots, D\}$ can be indexed by

row and column $i_d, j_d \in \{1, \ldots, m_d\}$ which again yields a submatrix defined either as $\boldsymbol{G}^{(d-1)}_{i_d+j_d-1}$ or $\boldsymbol{G}^{(d-1)}_{m_d+i_d-j_d}$ (depending on whether the $d^{\text{th}}$-level has Hankel structure as in Eq. (10) or Toeplitz structure as in Eq. (11)). Each entry in a block Hankel–Toeplitz matrix can also be expressed by the entry $\boldsymbol{\gamma}_{k_1,k_2,\ldots,k_D}$ of a tensor $\boldsymbol{\gamma} \in \mathbb{R}^{K_1 \times K_2 \times \ldots \times K_D}$, where $K_d = 2m_d - 1$, and

$$k_d = \begin{cases} i_d + j_d - 1, & \text{if level } d \text{ is Hankel} \\ m_d + i_d - j_d, & \text{if level } d \text{ is Toeplitz.} \end{cases} \tag{12}$$

A crucial property of the block Hankel–Toeplitz structure is the preservation of structure under addition. Assume that $\boldsymbol{A}$ and $\boldsymbol{B}$ are two block Hankel–Toeplitz matrices and that they are structurally identical, in the sense that they have the same number of levels, the same number of entries in each block, and each level shares either Toeplitz or Hankel properties. Then, let each entry of $\boldsymbol{A}$ and $\boldsymbol{B}$ be given by $\alpha_{k_1,k_2,\ldots,k_D}$ for a tensor $\alpha \in \mathbb{R}^{m_1,\ldots,m_D}$ and $\beta_{k_1,k_2,\ldots,k_D}$ for $\beta \in \mathbb{R}^{m_1,\ldots,m_D}$, respectively, with $k_d$ defined in Eq. (15). Each entry in $\boldsymbol{A} + \boldsymbol{B}$ is then given by the sum of the entries $\alpha_{k_1,\ldots,k_D} + \beta_{k_1,\ldots,k_D}$. Thus, the sum of two block Hankel–Toeplitz matrices with identical structure is also a block Hankel–Toeplitz matrix. By associativity of matrix addition, a sum $\sum_{n=1}^{N} \boldsymbol{G}_n$ of $N$ Hankel–Toeplitz matrices $\{\boldsymbol{G}_1, \ldots, \boldsymbol{G}_N\}$ with identical structure is therefore itself a Hankel–Toeplitz matrix.

### 3.3 Kronecker Products of Hankel–Toeplitz Matrices and Block Hankel–Toeplitz Matrices

A special case of a class of matrices $\boldsymbol{G}$ which has block Hankel–Toeplitz structure are Kronecker products of $D$ Hankel and Toeplitz matrices $\{\boldsymbol{G}^{(1)}, \ldots, \boldsymbol{G}^{(D)}\}$, i.e.,

$$\boldsymbol{G} = \bigotimes_{d=1}^{D} \boldsymbol{G}^{(d)} := \boldsymbol{G}^{(1)} \otimes \boldsymbol{G}^{(2)} \otimes \cdots \otimes \boldsymbol{G}^{(D)}. \tag{13}$$

An equivalent definition of the Kronecker product gives each entry on the $i^{\text{th}}$ row and $j^{\text{th}}$ column in the block Hankel–Toeplitz matrix $\boldsymbol{G}$ as an expression of the entries on the $i_d^{\text{th}}$ row and $j_d^{\text{th}}$ column of each matrix $\boldsymbol{G}^{(d)}$ according to

$$\boldsymbol{G}_{i,j} = \prod_{d=1}^{D} \boldsymbol{G}^{(d)}_{i_d,j_d}. \tag{14}$$

Note that there is a one-to-one map between each index $i, j$ and the index sets $\{i_1, \ldots, i_D\}$ and $\{j_1, \ldots, j_D\}$. As each matrix $\boldsymbol{G}^{(d)}$ has Hankel or Toeplitz structure, the entries can equivalently be defined by a vector $\boldsymbol{\gamma}^{(d)}$ with $2m_d - 1$ entries. Each entry $\boldsymbol{G}_{i,j}$ is therefore given by

$$\boldsymbol{G}_{i,j} = \prod_{d=1}^{D} \boldsymbol{\gamma}^{(d)}_{k_d} = \boldsymbol{\gamma}_{k_1,\ldots,k_D}, \tag{15}$$

where $k_d$ is defined in Eq. (12), and where $\boldsymbol{\gamma} := \bigotimes_{d=1}^{D} \boldsymbol{\gamma}^{(d)}$ is a rank-1 tensor with $\prod_{d=1}^{D}(2m_d - 1)$ elements.

### 3.4 Main Results

We are now ready to state our main findings. The following two theorems rely on the product decomposition Eq. (5) and study each dimension $d$ separately, as is evidently possible from Eq. (7). Our first theorem generalizes a result by Greengard et al. (2023) regarding complex exponential basis functions. Our first theorem establishes that if the product $\boldsymbol{\phi}^{(d)}(x_n^{(d)})[\boldsymbol{\phi}^{(d)}(x_n^{(d)})]^\top$ has Hankel or Toeplitz structure, the resulting precision matrix only has $\prod_{d=1}^{D}(2m_d - 1)$ unique entries, reducing the computational complexity of instantiating it from $\mathcal{O}(NM^2)$ to $\mathcal{O}(NM)$. We formalize this in the following theorem.

**Theorem 3.1.** *If the matrix*

$$\boldsymbol{G}^{(d)}(x_n^{(d)}) := \boldsymbol{\phi}^{(d)}(x^{(d)})\big[\boldsymbol{\phi}^{(d)}(x^{(d)})\big]^\top, \tag{16}$$

*is a Hankel or Toeplitz matrix for all $x^{(d)} \in \mathbb{R}$ along each dimension $d$, the information matrix $\boldsymbol{\Phi}^\top \boldsymbol{\Phi}$ will be a multi-level block Hankel or Toeplitz matrix, and therefore have $\prod_{d=1}^{D}(2m_d - 1)$ unique entries.*

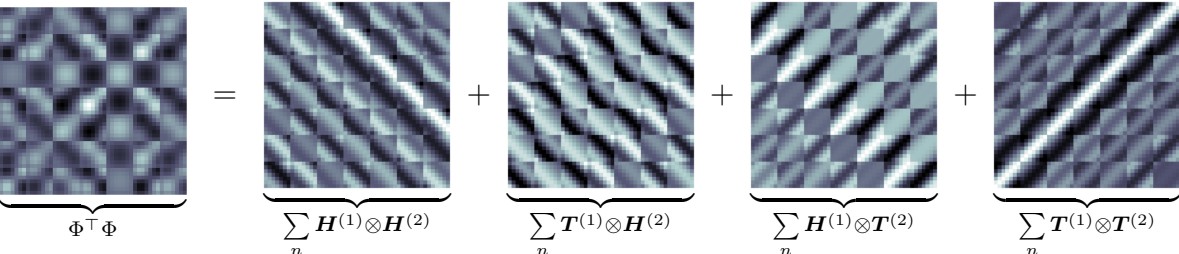

Figure 4: The precision matrix for sinusoidal BFs in two dimensions has neither Hankel nor Toeplitz structure. However, it can be decomposed into $2^D = 4$ matrices, which each have block Hankel–Toeplitz structure. Here, 7 BFs are placed along each of the two dimensions, giving a total of 49 BFs.

*Proof.* Assume that the matrix $\boldsymbol{G}^{(d)}(x_n^{(d)}) := \boldsymbol{\phi}^{(d)}(x_n^{(d)})[\boldsymbol{\phi}^{(d)}(x_n^{(d)})]^\top$ is Hankel or Toeplitz. The precision matrix can then be expressed as

$$\boldsymbol{\Phi}^\top\boldsymbol{\Phi} = \sum_{n=1}^N \otimes_{d=1}^D \boldsymbol{\phi}^{(d)}(x_n^{(d)}) \left[\boldsymbol{\phi}^{(d)}(x_n^{(d)})\right]^\top = \sum_{n=1}^N \otimes_{d=1}^D \boldsymbol{G}^{(d)}(x_n^{(d)}), \tag{17}$$

where the matrix $\otimes_{d=1}^D \boldsymbol{G}^{(d)}(x_n^{(d)})$ is multi-level Hankel or Toeplitz by definition, see Sec. 3.3. Further, the sum of several D-level block Hankel–Toeplitz matrices is itself a $D$-level block Hankel–Toeplitz matrix, see Sec. 3.2. Since each matrix $\boldsymbol{G}^{(d)}(x_n^{(d)})$ has at most $(2m_d - 1)$ unique entries, the matrix $\boldsymbol{\Phi}^\top\boldsymbol{\Phi} = \sum_{n=1}^N \otimes_{d=1}^D \boldsymbol{G}^{(d)}(x_n^{(d)})$ therefore has at most $M = \prod_{d=1}^D (2m_d - 1)$ unique entries. $\square$

The preceding theorem holds true for, for instance, polynomial and complex exponential BFs, which we establish in Corollaries 3.2 and 3.3.

**Corollary 3.2.** *The precision matrix for polynomial BFs defined by*

$$\phi_{i_d}^{(d)}(x^{(d)}) = (x^{(d)})^{i_d - 1}, \tag{18}$$

*can be represented by a tensor with $\prod_{d=1}^D 2m_d - 1$ entries.*

*Proof.* See App. A for a proof. $\square$

**Corollary 3.3.** *The precision matrix for complex exponential BFs defined by*

$$\phi_j^C(x) = \exp(i\pi j^\top x) = \prod_{d=1}^D \exp(i\pi j_d x_d), \tag{19}$$

*can be represented by a tensor with $\prod_{d=1}^D 2m_d - 1$ entries.*

*Proof.* See App. B for a proof. $\square$

For some BFs, the structure of the product $\boldsymbol{\phi}^{(d)}(x_n^{(d)})[\boldsymbol{\phi}^{(d)}(x_n^{(d)})]^\top$ is more intricate, but is still of a favorable, exploitable nature. This is clearly evident from Fig. 4, where the precision matrix for sinusoidal BFs in two dimensions is visualized. In particular, for some BFs, the product is the sum of a Hankel and a Toeplitz matrix, such that the precision matrix only has $\prod_{d=1}^D 3m_d$ unique entries, again reducing the computational cost of computing it to $\mathcal{O}(NM)$. We formalize this in the following theorem.

**Theorem 3.4.** *If the product $\boldsymbol{\phi}^{(d)}(x_n^{(d)})[\boldsymbol{\phi}^{(d)}(x_n^{(d)})]^\top$ is the sum of a Hankel matrix denoted $\boldsymbol{G}^{(d),(1)}(x_n^{(d)})$ and a Toeplitz matrix $\boldsymbol{G}^{(d),(-1)}(x_n^{(d)})$, and there exists a function $g^{(d)}(k_d, x_n^{(d)})$ such that $\boldsymbol{G}_{i_d,j_d}^{(d),(1)}(x_n^{(d)}) = g(i_d + j_d, x_n^{(d)})$ and $\boldsymbol{G}_{i_d,j_d}^{(d),(-1)}(x_n^{(d)}) = -g(i_d - j_d, x_n^{(d)})$, all entries in the precision matrix can be represented by a tensor $\boldsymbol{\gamma}(k_1, k_2, \ldots, k_D)$ with $\prod_{d=1}^D 3m_d$ entries.*

*Proof.* The precision matrix can in this case be expressed as

$$
\overbrace{\boldsymbol{\Phi}^\top \boldsymbol{\Phi}}^{:=\boldsymbol{C}} = \sum_{n=1}^N \otimes_{d=1}^D \boldsymbol{\phi}^{(d)}(x_n^{(d)}) \left[\boldsymbol{\phi}^{(d)}(x_n^{(d)})\right]^\top
$$
$$
= \sum_{n=1}^N \otimes_{d=1}^D \left(\boldsymbol{G}^{(d),(1)}(x_n^{(d)}) + \boldsymbol{G}^{(d),(-1)}(x_n^{(d)})\right)
$$
$$
= \sum_{p=1}^{2^D} \left(\prod_{d=1}^D e_p^{(d)}\right) \sum_{n=1}^N \otimes_{d=1}^D \boldsymbol{G}^{(d),(e_p^{(d)})}(x_n^{(d)}), \tag{20}
$$

where $e_p = \{e_p^{(1)}, \ldots, e_p^{(D)}\} \in S^D$ and $S^D = \{1, -1\}^D$ is a set containing $2^D$ elements. Each of the $2^D$ matrices $\left(\prod_{d=1}^D e_p^{(d)}\right) \sum_{n=1}^N \otimes_{d=1}^D \boldsymbol{G}^{(d),(e_p^{(d)})}(x_n^{(d)})$ is now the Kronecker product between $D$ Hankel or Toeplitz matrices. The entries of $\boldsymbol{C}$ can be expressed element-wise as

$$
\boldsymbol{C}_{i,j} = \sum_{p=1}^{2^D} \left(\prod_{d=1}^D e_p^{(d)}\right) \sum_{n=1}^N \prod_{d=1}^D g(i_d + e_p^{(d)} j_d, x_n^{(d)}). \tag{21}
$$

If we define a tensor $\boldsymbol{\gamma}$ as

$$
\boldsymbol{\gamma}_{k_1,\ldots,k_D} = \sum_{n=1}^N \prod_{d=1}^D g(k_d, x_n^{(d)}) \tag{22}
$$

for indices $k_d = 1 - m_d, 2 - m_d, \ldots, 2m_d - 1, 2m_d$, each entry of the precision matrix $\boldsymbol{C}_{i,j}$ can be expressed as

$$
\boldsymbol{C}_{i,j} = \sum_{p=1}^{2^D} \left(\prod_{d=1}^D e_p^{(d)}\right) \boldsymbol{\gamma}_{i_1 + e_p^{(1)} j_1, \ldots, i_D + e_p^{(D)} j_D}. \tag{23}
$$

As each sum $k_d = i_d + e_p^{(d)} j_d$ is an integer between $1 - m_d$ and $2m_d$, the tensor $\boldsymbol{\gamma}$ will have $\prod_{d=1}^D 3m_d$ entries. $\qquad\square$

The preceding theorem applies to, for instance, the BFs in an HGP (Solin & Särkkä, 2020) defined on a rectangular domain, formalized in Corollary 3.5. It further holds for multiple other works using similar BFs, formalized in Corollary 3.6. We remark that for $D = 1$, we require $m_d > 3$ for any savings to take effect, whereas for $D > 1$, $m_d \geq 2$ suffices.

**Corollary 3.5.** *The precision matrix in an HGP defined on a rectangular domain $[-L_1, L_1] \times \cdots \times [-L_D, L_D]$ can be represented by a tensor with $\prod_{d=1}^D 3m_d$ entries.*

*Proof.* See App. C for a proof. $\qquad\square$

**Corollary 3.6.** *The precision matrix in a GP approximated by sinusoidal and cosine BFs with frequencies on a grid (such as the regular Fourier features described in Hensman et al. (2017) and Wahls et al. (2014), the Fourier approximations to periodic kernels described in Tompkins & Ramos (2018), the quadrature Fourier features described in Mutný & Krause (2018), the equispaced-version of sparse spectrum BFs described in Lázaro-Gredilla et al. (2010), or the one-dimensional special case of Dutordoir et al. (2020)) can be represented by a tensor with $\prod_{d=1}^D 3m_d$ entries.*

*Proof.* See App. D for a proof. $\qquad\square$

### 3.5 Outlook and Practical Use

Both Theorems 3.1 and 3.4 reduce the computational complexity of calculating the entries of the precision matrix Eq. (7) from $\mathcal{O}(NM^2)$ to $\mathcal{O}(NM)$. This enables us to scale the number of BFs significantly more than previously, before running into computational, or storage related, problems. For clarity, the standard HGP is given in Alg. 1 with the original approach in red and our proposed approach in blue. As compared to the standard (offline) HGP, the only change we make is the computation of the precision matrix.

We remark that computing the posterior mean and variance of the BF expansion now costs $\mathcal{O}(NM + M^3)$, now possibly dominated by the $M^3$ term. This cost also appears in the hyperparameter optimization, as we choose to optimize the MLL as in Solin & Särkkä (2020). Remedies for this cost are out of scope of this

---

**Algorithm 1** Sketch of an algorithm for Hilbert GP learning and inference. The original approach by Solin & Särkkä (2020) in red, our proposed approach in blue.

---

**Input:** Data as input–output pairs $\{(\boldsymbol{x}_i, y_i)\}_{i=1}^N$,
test inputs $\boldsymbol{x}_\star$, number of basis functions $M$

Compute $\boldsymbol{\Phi}^\top \boldsymbol{\Phi}$ at cost $\mathcal{O}(NM^2)$     ▷ Eq. (7)
Compute $\boldsymbol{\gamma}$ at cost $\mathcal{O}(NM)$     ▷ Eq. (22)
Construct $\boldsymbol{\Phi}^\top \boldsymbol{\Phi}$ using $\boldsymbol{\gamma}$ at cost $\mathcal{O}(M^2)$
**repeat**
     Optimize *Marginal Log-Likelihood* (MLL) w.r.t. hyperparameters at cost $\mathcal{O}(M^3)$
**until** Convergence
Perform GP inference using the pre-calculated matrices. This entails computing the posterior mean and covariance, at a computational cost of $\mathcal{O}(M^3)$.

---

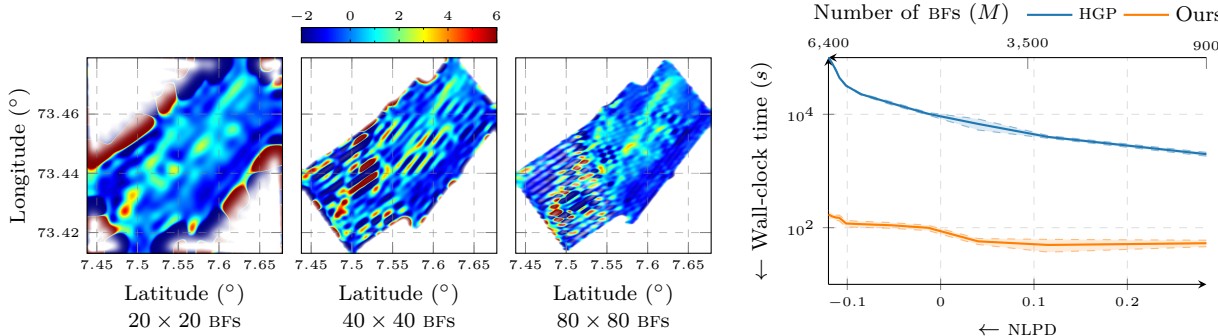

(a) Predictive means indicated by color intensity with transparency proportional to the predictive variance. Increasing number of BFs from left to right.

(b) Wall-clock time to compute precision matrix by sequentially including each data point over NLPD.

Figure 5: Our proposed computational scheme reduces the computation time for datasets with high-frequency variations, as these require many BFs to achieve accurate reconstruction. This underwater magnetic field has lower *Negative Log Predictive Density* (NLPD) with a large amount (6400) compared to a smaller amount (400) of BFs. For 6400 BFs, our computational scheme reduced the required time to compute the precision matrix from 2.7 hours to 1.7 minutes.

paper, but could potentially be reduced through the use of efficient approximate matrix inverses and trace estimators (see, e.g., Davies, 2015). Another consequence of these theorems is that multi-agent systems that collaborate to learn the precision matrix such as Jang et al. (2020), Viset et al. (2023) or Pillonetto et al. (2019) can do this by communicating $\mathcal{O}(M)$ bits instead of $\mathcal{O}(M^2)$ bits.

## 4 Experiments

We demonstrate the storage and computational savings of our structure exploiting scheme by means of three numerical experiments. The experiments demonstrate the practical efficiency of our scheme using the HGP. We reiterate that the savings are *without* additional approximations and the posterior is therefore *exactly* equal to that of the standard HGP. Further, as HGPs adhere to Theorem 3.4, this demonstration is a representative example of the speedups that can be expected using, e.g., regular Fourier features (Hensman et al., 2017), or BF expansions of periodic kernels (Dutordoir et al., 2020). Since Theorem 3.1 requires computing and storing only $2^D M$ components whereas Theorem 3.4 requires $3^D M$, our experiments demonstrate a practical upper bound on the storage and computational savings, a "worst case".

Our first experiment demonstrates the computational scaling of our scheme on a simulated 3D dataset. Secondly, we consider a magnetic field mapping example with data collected by an underwater vessel, as an application where the high-frequency content of the data requires a large amount of BFs to reconstruct the

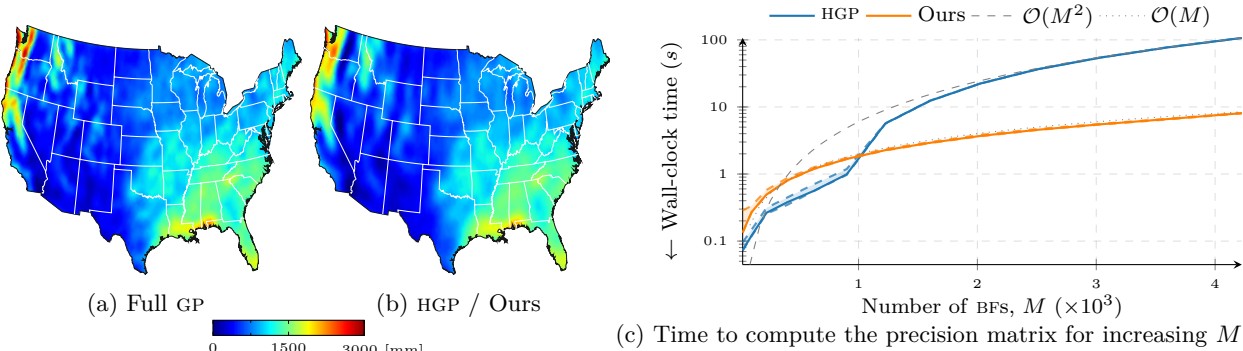

(a) Full GP  (b) HGP / Ours  (c) Time to compute the precision matrix for increasing $M$

Figure 6: These experiments recover the results from Solin & Särkkä (2020) exactly for predicting yearly precipitation levels across the US, and measure the wall-clock time needed by our proposed computational scheme. The HGP efficiently approximates the full GP solution using $m_1 = m_2 = 45$, totaling $M = 2025$ BFs.

field. Thirdly, a precipitation dataset is used, mirroring an example in Solin & Särkkä (2020), improving the computational scaling in that particular application even further than the standard HGP. Our freely available HGP reference implementation along with hyperparameter optimization is written for GPJax (Pinder & Dodd, 2022). All timing experiments are run on an HP Elitebook 840 G5 laptop (Intel i7-8550U CPU, 16GB RAM). For fair comparison, we naively loop over data points, to avoid any possible low-level optimization skewing the results.

**Computational Scaling**  We compare the time necessary for computing the precision matrix for $N = 500$ data points for the standard HGP as well as for our structure exploiting scheme. The results are presented in Fig. 1 for an increasing number of BFs. After $M = 14000$, the HGP is no longer feasible due to memory constraints, whereas our formulation scales well above that, but stop at $M = 64000$ for clarity in the illustration. Clearly, the structure exploitation gives a significantly lower computational cost than the standard HGP, even for small quantities of BFs. Further, it drastically reduces the memory requirements, where for $M = 14000$, the HGP requires roughly 1.5 GB for storing the precision matrix, while our formulation requires roughly 2.8 MB using 64-bit floats. This makes it possible for us to scale the number of BFs significantly more before running into computational or storage restrictions. It is noteworthy that even though the implementation is in a high-level language, we still see significant computational savings.

**Magnetic Field Mapping**  As HGPs are commonly used to model and estimate the magnetic field for, e.g., mapping purposes (Solin et al., 2015; Kok & Solin, 2018), we consider a magnetic field mapping example and demonstrate the ability of our computational scheme to scale HGPs to spatially vast datasets. The data was gathered sequentially in a lawn-mower path in a region approximately $7 \times 7$ km large by an underwater vessel outside the coast of Norway ($d = 2$ and $N = 1.39$ million). The data was split into a training set and test set with roughly a 50/50 split, deterministically split by a grid pattern, to ensure reasonable predictions in the entire data domain, see App. F.2 for more details. We vary the amount of BFs and compare the time required to sequentially include each new data point in the precision matrix as well as the *Negative Log Predictive Density* (NLPD). As the underwater magnetic field covers a large area, a large number of BFs are required to accurately represent the field, see Fig. 5a where the details of the predicted magnetic field is captured more accurately for an increasing number of BFs. This is also apparent from the decreasing NLPD as the number of BFs increases, see Fig. 5b. At 6400 BFs, the necessary computation time is several orders of magnitude lower for our approach compared to the standard HGP.

**U.S. Precipitation Data**  We consider a standard precipitation data set containing US annual precipitation summaries for year 1995 ($d = 2$ and $N = 5776$) (Vanhatalo & Vehtari, 2008). We exactly mimic the setup in Solin & Särkkä (2020) and primarily focus our evaluation on the calculation of the precision matrix. The time for computing the precision matrix is visualized in Fig. 6c, where our approach clearly outperforms the standard HGP. The predictions on a dense grid over the continental US can be found in Figs. 6a and 6b, where the HGP manages to capture both the large-scale as well as the small-scale variations well.

## 5 Related Work

The computational scheme that we detail here can be used to speed up a range of approximate GP and kernel methods with BFs that satisfy the Hankel–Toeplitz structure we use (see e.g. Tompkins & Ramos, 2018; Solin & Särkkä, 2020; Hensman et al., 2017). It is also applicable to Lázaro-Gredilla et al. (2010) in the special case where the considered frequencies of the BFs are equidistant, as well as Dutordoir et al. (2020) when the input space is one-dimensional. However, there is also a wide range of GP approximations that do not have the Hankel–Toeplitz structure required for Theorem 3.1 or Theorem 3.4 to apply. While the structure we exploit in Theorem 3.4 is apparent in the quadrature Fourier feature approach of Mutný & Krause (2018) when the frequencies are on a structured Cartesian grid, other quadrature-like methods are not possible to speed-up in similar ways. The most well-known method is the random Fourier feature approach (Rahimi & Recht, 2007), where speed-up is not possible as the frequencies are sampled at random. Similarly, the Gaussian quadrature approaches of Dao et al. (2017); Shustin & Avron (2022) and the random quadrature approach of Munkhoeva et al. (2018) do not constrain frequencies to a regular grid and therefore do not have the Hankel–Toeplitz structure required by Theorem 3.4.

As was pointed out by Quiñonero-Candela & Rasmussen (2005), inducing point approaches can also be viewed as BF approximations. The inducing variable approaches essentially summarize the data in a set of inducing variables (Snelson & Ghahramani, 2005; Seeger et al., 2003; Csató & Opper, 2002), which most commonly represent function values at a certain set of inputs, even though other choices are possible (Lázaro-Gredilla & Figueiras-Vidal, 2009). Another closely related approach which also uses inducing points, with similar computational complexity of $\mathcal{O}(NM^2)$, is the variational GP pioneered by Titsias (2009). In light of the BF viewpoint, all the aforementioned approaches therefore also involve computing a precision matrix, but even for ordered inducing points on a grid, the particular Hankel–Toeplitz structure we exploit does not exist in general.

A range of related work stacks additional approximations on top of the sparse basis function/inducing point approximations (Yadav et al., 2021; Izmailov et al., 2018; Hensman et al., 2013). The purpose of these additional approximations is typically to either reduce the computational complexity of computing the precision matrix for example by using BFs with compact support, or by implementing an approximate algorithm for inverting the precision matrix. A well-known structure exploiting method is *Structured Kernel Interpolation* (SKI) (Wilson & Nickisch, 2015; Yadav et al., 2021; Izmailov et al., 2018), which approximates the precision matrix through cubic interpolation between inducing points on a regular grid. An alternative approach that improves the computational complexity of the variational GP, is the *Stochastic Variational GP* (SVGP) (Hensman et al., 2013). The SVGP is the *de facto* standard approach for large–scale GPs, due to the possibility of utilizing mini-batching for training, greatly speeding up hyperparameter (and variational parameter) learning (Hensman et al., 2013; 2015), with implementation available in, e.g., GPyTorch and GPflow (Gardner et al., 2018; de G. Matthews et al., 2017).

While this paper focuses on BF approximations to GPs, there is a lot of work on approximating exact GP regression through means of, e.g., *Conjugate Gradient* (CG) descent (Gibbs & MacKay, 1996; Artemev et al., 2021; Gardner et al., 2018). These methods view GP regression as the solution to a linear system of equations and seek to solve this approximately. This typically reduces the cost of exact GP regression from $\mathcal{O}(N^3)$ to $\mathcal{O}(IN^2)$, where $I$ is the number of CG iterations. Davies (2015) uses CG as the driving scalability mechanism, but further reduces the computational complexity to $\mathcal{O}(NMI)$ through use of *M-efficient kernels*, which essentially constitute the approaches from Quiñonero-Candela & Rasmussen (2005) as well as compact kernels (see, e.g., Kullberg et al., 2021; Buhmann, 2003; Wu, 1995). CG has also been combined with SKI in the KISS-GP framework of Wilson & Nickisch (2015) to reduce the computational complexity of (mean) inference to $\mathcal{O}(N + M \log M)$. Pleiss et al. (2018) extended KISS-GP with the Lanczos algorithm to reduce the complexity of computing the predictive variance to $\mathcal{O}(k)$ after pre-computation, where $k$ is the number of Lanczos iterations. Recently, *Stochastic Gradient Descent* (SGD) was introduced as an alternative to CG (Lin et al., 2023a;b), potentially with better performance for ill-conditioned datasets. Technically, any of these ideas are straightforward to include in the HGP as well and could potentially be used to reduce the $\mathcal{O}(M^3)$ complexity of inference and hyperparameter learning.

Other previous work has discovered that functions of difference matrices (a special case of Toeplitz matrices) are also sums of Hankel and Toeplitz matrices in the one-dimensional case, and Kronecker products of these

in the two-dimensional case (Strang & MacNamara, 2014). Since our matrices are also Kronecker products of Hankel and Toeplitz matrices, we end up exploiting similar structures as Strang & MacNamara (2014) arising in a different situation.

## 6 Conclusion

Our contribution details a computational approach for exploiting Hankel and Toeplitz structures that appear in multiple BF approximation schemes to kernels for GPs. These structures allow us to reduce the computational complexity of computing the corresponding precision matrix from $\mathcal{O}(NM^2)$ to $\mathcal{O}(NM)$ *without* further approximations. Further, our approach reduces the storage requirement for containing all necessary information about the posterior to make predictions from $\mathcal{O}(M^2)$ to $\mathcal{O}(M)$. The Hankel and Toeplitz structures appear because of the particular BFs that are used to approximate the kernel, *not* the kernel itself. The reduced computational and storage requirements are particularly beneficial in the HGP where more BFs allow us to capture higher frequencies of the kernel spectrum, otherwise unattainable without significant computational resources. We foresee that our contribution will allow HGPs to tackle larger problems without the need for extensive specialized hardware, opening up approximate GP learning and inference for a wider audience. Future work could investigate if the results can be generalized to wider ranges of BFs, which is easily verified through Theorems 3.1 and 3.4. Another potential is to investigate ways of approximately decomposing an already known precision matrix into Hankel–Toeplitz matrices which does not admit an analytical such decomposition. This would yield the same computational benefits but with some potential loss of accuracy.

A reference implementation built on top of GPJax is available at https://github.com/AOKullberg/hgp-hankel-structure.

## Impact Statement

This work develops numerical methods for the wide field of machine learning, where the goal is to make existing methods more compute-efficient and open new avenues for developing future methods. There are many potential uses and therefore societal consequences of such methods, none of which we see the need to specifically highlight here.

## Acknowledgements

We would like to express our gratitude to the supervisors (Gustaf Hendeby, Isaac Skog, Kim Batselier, Manon Kok and Rudy Helmons) of the three PhD candidates (Anton Kullberg, Frederiek Wesel and Frida Viset) involved in this project. They secured the funding that made this research possible. Their support in providing the necessary resources and their encouragement for our development as independent researchers have been invaluable. Their contributions have thus indirectly shaped this work, and we are grateful for their continued guidance and support. We would like to thank the anonymous reviewers for their numerous suggestions which have greatly improved the quality of this paper. Frederiek Wesel, and thereby this work, is supported by the Delft University of Technology AI Labs program. Arno Solin acknowledges funding from the Research Council of Finland (grant id 339730). The underwater magnetic field data used were collected by MARMINE/NTNU research cruise funded by the Research Council of Norway (Norges Forskningsråd, NFR) Project No. 247626/O30 and associated industrial partners. Ocean Floor Geophysics provided the magnetometer that was used for magnetic data acquisition and pre-processed the magnetic data. The authors declare no competing interests.

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

## Appendix

The majoriy of the appendix covers proofs of the corollaries following from Theorems 3.1 and 3.4. The rest is dedicated to further details on our empirical experiments as well as visual explanations of the structures that are exploited in the main body of the paper. The appendix is organized as follows. App. A proves Theorem 3.1 for polynomial *Basis Functions* (BFs). App. B proves Theorem 3.1 for complex exponential BFs. App. C proves Theorem 3.4 for Hilbert space BFs defined on a rectangular domain. App. D proves Theorem 3.4 for ordinary Fourier features. App. E contains visual representations of the structures explained in the main body of the paper. Lastly, App. F provides a full description of all the included experiments and the data used in them, with additional plots and results.

## A  Proof of Corollary 3.2 (use of Theorem 3.1 for Polynomial Basis Functions)

Polynomial BFss (as used in for example Chen et al. (2018)) are defined along each dimension as

$$\phi_{i_d}^{(d)}(x^{(d)}) = (x^{(d)})^{i_d - 1} \tag{24}$$

By selecting $g_{i_d + j_d}(x^{(d)}) = (x^{(d)})^{j_d + i_d - 1}$, the product of two component BFs becomes

$$\phi_{i_d}^{(d)}(x^{(d)})\phi_{j_d}^{(d)}(x^{(d)}) = (x^{(d)})^{i_d - 1}(x^{(d)})^{j_d - 1} = (x^{(d)})^{j_d + i_d - 1} = g_{i_d + j_d}, \tag{25}$$

which shows that the conditions for Theorem 3.1 is satisfied.

## B  Proof of Corollary 3.3 (use of Theorem 3.1 for Complex Exponential Basis Functions)

Complex exponential BFs (as used in for example Novikov et al. (2018)) are defined along each dimension as

$$\phi_{i_d}^{(d)}(x^{(d)}) = \exp(i\pi j_d x^{(d)}) \tag{26}$$

By selecting $g_{i_d + j_d}(x^{(d)}) = \exp(i\pi (j_d + i_d) x^{(d)})$, the product of two component BFs becomes

$$\phi_{i_d}^{(d)}(x^{(d)})\phi_{j_d}^{(d)}(x^{(d)}) = \exp(i\pi i_d x^{(d)})\exp(i\pi j_d x^{(d)}) = \exp(i\pi (j_d + i_d) x^{(d)}) = g_{i_d + j_d}, \tag{27}$$

which shows that the conditions for Theorem 3.1 is satisfied.

## C  Proof of Corollary 3.5 (use of Theorem 3.4 for Hilbert Space Basis Functions on a Rectangular Domain)

Hilbert space BFs on a rectangular domain are defined according to (Solin & Särkkä, 2020)

$$\phi_i(x) = \prod_{d=1}^{D} \frac{1}{\sqrt{L_d}} \sin\left(\frac{\pi i_d(x_d + L_d)}{2L_d}\right) = \prod_{d=1}^{D} \frac{1}{\sqrt{L_d}} \cos\left(\frac{\pi i_d(x_d + L_d)}{2L_d} - \frac{\pi}{2}\right), \tag{28}$$

where the indices $i = 1, \ldots, M^D$ has a one-to-one mapping with all possible combinations of the indices $i_1, i_2, \ldots, i_D$, given that each index $i_d = \{1, \ldots, D\}$.

This corresponds to defining the BFs according to Eq. (6a), with each entry of $\boldsymbol{\phi}^{(d)}(x)$ defined as

$$\phi_{i_d}^{(d)}(x) = \frac{1}{\sqrt{L_d}} \sin\left(\frac{\pi i_d(x + L_d)}{2L}\right) = \frac{1}{\sqrt{L_d}} \cos\left(\frac{\pi i_d(x + L_d)}{2L_d} - \frac{\pi}{2}\right),$$

where $i_d \in \{1, \ldots, m\}$. Define a linear function $\theta_{i_d} : \mathbb{R} \to \mathbb{R}$ as

$$\theta_{i_d}(x) = \frac{\pi i_d(x + L_d)}{2L_d} - \frac{\pi}{2}.$$

Hence, the BFs can be written as

$$\boldsymbol{\phi}^{(d)}(x) = \frac{1}{\sqrt{L_d}} \begin{bmatrix} \cos(\theta_1) \\ \vdots \\ \cos(\theta_M) \end{bmatrix}.$$

The product $\boldsymbol{\phi}^{(d)}(x_n^{(d)})\boldsymbol{\phi}^{(d)}(x_n^{(d)})^\top$ is then given by

$$\boldsymbol{\phi}^{(d)}(x_n^{(d)})\boldsymbol{\phi}^{(d)}(x_n^{(d)})^\top = \frac{1}{L_d} \sum_{n=1}^{N} \begin{bmatrix} \cos(\theta_1)\cos(\theta_1) & \dots & \cos(\theta_1)\cos(\theta_M) \\ \vdots & \ddots & \vdots \\ \cos(\theta_M)\cos(\theta_M) & \dots & \cos(\theta_M)\cos(\theta_M) \end{bmatrix}.$$

Then, note that

$$\cos(u)\cos(v) = \frac{\cos(u+v) + \cos(u-v)}{2}. \tag{29}$$

When we apply this to each entry in the matrix, we get that

$$\boldsymbol{\phi}^{(d)}(x_n^{(d)})\boldsymbol{\phi}^{(d)}(x_n^{(d)})^\top =$$
$$\frac{1}{2L_d}\left( \underbrace{\sum_{n=1}^{N} \begin{bmatrix} \cos(\theta_1+\theta_1) & \dots & \cos(\theta_1+\theta_M) \\ \vdots & \ddots & \vdots \\ \cos(\theta_M+\theta_1) & \dots & \cos(\theta_M+\theta_M) \end{bmatrix}}_{\triangleq \boldsymbol{G}^{d,(1)}} + \underbrace{\sum_{n=1}^{N} \begin{bmatrix} \cos(\theta_1-\theta_1) & \dots & \cos(\theta_1-\theta_M) \\ \vdots & \ddots & \vdots \\ \cos(\theta_M-\theta_1) & \dots & \cos(\theta_M-\theta_M) \end{bmatrix}}_{\triangleq \boldsymbol{G}^{d,(-1)}} \right),$$

where $\boldsymbol{G}^{d,(1)}$ and $\boldsymbol{G}^{d,(-1)}$ are a Hankel and a Toeplitz matrix, respectively. Then, since $\cos(\theta_{i_d} \pm \theta_{j_d}) = \pm\sin(\theta_{i_d \pm j_d}) \triangleq g(i_d \pm j_d) = g(k_d)$, there exists a function $g(k_d)$ that fulfills conditions for Theorem 3.4.

## D  Proof of Corollary 3.6 (use of Theorem 3.4 for Ordinary Fourier Features)

Ordinary Fourier features for separable kernels are defined slightly differently from Hilbert Space BFs (Hensman et al., 2017), in that they consider both a sine and a cosine function for each considered frequency. The set of BFs are given by

$$\begin{aligned} \boldsymbol{\phi}^{(d)}(x) &= \begin{bmatrix} \boldsymbol{\phi}_{\sin}^{(d)}(x) & \boldsymbol{\phi}_{\cos}^{(d)}(x) \end{bmatrix} \\ &= \begin{bmatrix} \sin(\Delta x) & \sin(2\Delta x) & \dots & \sin(m_d\Delta) & \cos(\Delta x) & \cos(2\Delta x) & \dots & \cos(m_d\Delta) \end{bmatrix}, \end{aligned} \tag{30}$$

where $\Delta$ determines the spacing of the Fourier features in the frequency domain. The precision matrix can therefore be expressed as

$$\boldsymbol{\Phi}^\top\boldsymbol{\Phi} = \begin{bmatrix} \boldsymbol{\Phi}_{\sin}^\top\boldsymbol{\Phi}_{\sin} & (\boldsymbol{\Phi}_{\cos}^\top\boldsymbol{\Phi}_{\sin})^\top \\ \boldsymbol{\Phi}_{\cos}^\top\boldsymbol{\Phi}_{\sin} & \boldsymbol{\Phi}_{\cos}^\top\boldsymbol{\Phi}_{\cos} \end{bmatrix}, \tag{31}$$

and we can apply Theorem 3.4 directly to entries $\boldsymbol{\Phi}_{\sin}^\top\boldsymbol{\Phi}_{\sin}$ and $\boldsymbol{\Phi}_{\sin}^\top\boldsymbol{\Phi}_{\cos}$ to prove that each of these have a block Hankel–Toeplitz structure.

For the matrix $\boldsymbol{\Phi}_{\sin}^\top\boldsymbol{\Phi}_{\sin}$, the product $\boldsymbol{\phi}_{\sin}(x_n^{(d)})\boldsymbol{\phi}_{\sin}(x_n^{(d)})^\top$ can be expanded as

$$\{\boldsymbol{\phi}_{\sin}(x_n^{(d)})\boldsymbol{\phi}_{\sin}(x_n^{(d)})^\top\}_{i,j} = \sin(i\Delta x^{(d)})\sin(j\Delta x^{(d)}) = \{\boldsymbol{G}^{(d),(-1)}\}_{i,j} + \{\boldsymbol{G}^{(d),(1)}\}_{i,j}, \tag{32}$$

where

$$\{\boldsymbol{G}^{(d),(1)}\}_{i,j} = \cos(i\Delta x^{(d)} + j\Delta x^{(d)}), \tag{33}$$

and

$$\{\boldsymbol{G}^{(d),(-1)}\}_{i,j} = \cos(\Delta x^{(d)} - \Delta x^{(d)}). \tag{34}$$

The entry at row $i$ and column $j$ of $\boldsymbol{G}^{(d),(1)}$ can therefore be defined by the function $g(i+j) = -\cos(i+j)\Delta x^{(d)}$, and the entry at row $i$ and column $j$ of $\boldsymbol{G}^{(d),(-1)}$ is given by $-g(i-j)$, satisfying the requirements for Theorem 3.4.

For the matrix $\boldsymbol{\Phi}_{\cos}^{\top}\boldsymbol{\Phi}_{\sin}$, the product $\boldsymbol{\phi}_{\cos}(x_n^{(d)})\boldsymbol{\phi}_{\sin}(x_n^{(d)})^{\top}$ can be expanded as

$$\{\boldsymbol{\phi}_{\cos}(x_n^{(d)})\boldsymbol{\phi}_{\sin}(x_n^{(d)})^{\top}\}_{i,j} = \cos(i\Delta x^{(d)})\sin(j\Delta x^{(d)}) = \{\boldsymbol{G}^{(d),(-1)}\}_{i,j} + \{\boldsymbol{G}^{(d),(1)}\}_{i,j};  \tag{35}$$

where

$$\{\boldsymbol{G}^{(d),(1)}\}_{i,j} = \sin(i\Delta x^{(d)} + j\Delta x^{(d)}),  \tag{36}$$

and

$$\{\boldsymbol{G}^{(d),(-1)}\}_{i,j} = \sin(i\Delta x^{(d)} - j\Delta x^{(d)})  \tag{37}$$

The entry at row $i$ and column $j$ of $\boldsymbol{G}^{(d),(1)}$ can therefore be defined by the function $g(i+j) = -\cos((i+j)\Delta x^{(d)})$, and the entry at row $i$ and column $j$ of $\boldsymbol{G}^{(d),(-1)}$ is given by $-g(i-j)$, satisfying the requirements for Theorem 3.4.

For the matrix $\boldsymbol{\Phi}_{\cos}^{\top}\boldsymbol{\Phi}_{\cos}$, the product $\boldsymbol{\phi}_{\cos}(x_n^{(d)})\boldsymbol{\phi}_{\cos}(x_n^{(d)})^{\top}$ can be expanded as

$$\{\boldsymbol{\phi}_{\cos}(x_n^{(d)})\boldsymbol{\phi}_{\cos}(x_n^{(d)})^{\top}\}_{i,j} = \cos(i\Delta x^{(d)})\cos(j\Delta x^{(d)}) = \{\boldsymbol{G}^{(d),(-1)}\}_{i,j} + \{\boldsymbol{G}^{(d),(1)}\}_{i,j};  \tag{38}$$

where

$$\{\boldsymbol{G}^{(d),(1)}\}_{i,j} = \cos(i\Delta x^{(d)} + j\Delta x^{(d)}),  \tag{39}$$

and

$$\{\boldsymbol{G}^{(d),(-1)}\}_{i,j} = \cos(i\Delta x^{(d)} - j\Delta x^{(d)}).  \tag{40}$$

The entry at row $i$ and column $j$ of $\boldsymbol{G}^{(d),(1)}$ can therefore be defined by the function $g(i+j) = \cos((i+j)\Delta x^{(d)})$, and the entry at row $i$ and column $j$ of $\boldsymbol{G}^{(d),(-1)}$ is given by $g(i-j)$. An important notion for this matrix which makes it different from $\boldsymbol{\Phi}_{\sin}\boldsymbol{\Phi}_{\sin}^{\top}$ and $\boldsymbol{\Phi}_{\cos}\boldsymbol{\Phi}_{\sin}^{\top}$ is that this does not exactly satisfy the criteria for Theorem 3.4. The difference is that for the criteria to be exactly satisfied, entry $\{\boldsymbol{G}^{(d),(-1)}\}_{i,j}$ would have to be equal to $-g(i-j)$ rather than $g(i-j)$. However, by applying the proof of Theorem 3.4, but now noticing that the entries of $\boldsymbol{C} = \boldsymbol{\Phi}_{\cos}^{\top}\boldsymbol{\Phi}_{\cos}$ can be expressed element-wise as

$$\boldsymbol{C}_{i,j} = \sum_{p=1}^{2^D} \sum_{n=1}^{N} \prod_{d=1}^{D} g(i_d + e_p^{(d)} j_d),  \tag{41}$$

which allows us to use the tensor $\gamma$ as defined in Eq. (22) to express each entry of $\boldsymbol{C}$ according to

$$\boldsymbol{C}_{i,j} = \sum_{p}^{2^D} \gamma_{i_1 + e_p^{(1)}, \ldots, i_D + e_p^{(D)}}.  \tag{42}$$

## E  Overview of Hankel, Toeplitz, and D-level Block Hankel–Toeplitz Matrix Structures

An overview of Hankel, Toeplitz and D-level Block Hankel–Toeplitz matrix structures are given in Table A1.

## F  Experiment Details

More specific details of the numerical experiments are provided here with additional visualizations and explanations.

### F.1  U.S. Precipitation Data

The precipitation data is two-dimensional with $N = 5776$ data points first considered in (Vanhatalo & Vehtari, 2008). We perform regression in the lat/lon domain and first center the data (but do not perform scaling) and use a simple squared-exponential kernel. We optimized the MLL using GPJax (Pinder & Dodd, 2022) for both the HGP as well as the standard GP for 100 iterations using Adam (Kingma & Adam, 2015). The hyperparameters of the kernel and likelihood were initialized as $l = 1, \sigma_{SE}^2 = 10$ and $\sigma_e = 1$, where $l$ is the lengthscale, $\sigma_{SE}^2$ is the kernel variance and $\sigma_e$ the noise standard deviation. Purely for visualization purposes, the inputs were projected to a local coordinate system given by CRS 5070 which are used for all of the plots. The original data is plotted in Fig. A7. The timing experiments were run using $m_d = 5, 10, \ldots, 65$ BFs along each dimension, totaling between $M = 25$ and $M = 4225$ BFs.

Table A1: An overview of the matrix structure and tensor representation for Hankel, Toeplitz, block Hankel, block Toeplitz and block Hankel matrices. The illustrations are examples of matrices with the property described in each row. The illustrations contain one square for each matrix entry, where the color of the square corresponds to the value.

| STRUCTURE | DEFINITION | VISUAL | DOMAIN |
|---|---|---|---|
| HANKEL | $$\boldsymbol{H} = \begin{bmatrix} \gamma_1 & \gamma_2 & \cdots & \gamma_m \\ \gamma_2 & \gamma_3 & \cdots & \gamma_{m+1} \\ \vdots & \vdots & \ddots & \vdots \\ \gamma_m & \gamma_{m+1} & \cdots & \gamma_{2m-1} \end{bmatrix}$$ |  | $\gamma \in \mathbb{R}^K$ |
| TOEPLITZ | $$\boldsymbol{T} = \begin{bmatrix} \gamma_m & \cdots & \gamma_2 & \gamma_1 \\ \gamma_{m+1} & \cdots & \gamma_3 & \gamma_2 \\ \vdots & \ddots & \vdots & \vdots \\ \gamma_{2m-1} & \cdots & \gamma_{m+1} & \gamma_m \end{bmatrix}$$ |  | $\gamma \in \mathbb{R}^K$ |
| D-LEVEL BLOCK HANKEL | $$\boldsymbol{H}^{(D)} = \begin{bmatrix} \boldsymbol{H}_1^{(D-1)} & \boldsymbol{H}_2^{(D-1)} & \cdots & \boldsymbol{H}_{m_D}^{(D-1)} \\ \boldsymbol{H}_2^{(D-1)} & \boldsymbol{H}_3^{(D-1)} & \cdots & \boldsymbol{H}_{m_D+1}^{(D-1)} \\ \vdots & \vdots & \ddots & \vdots \\ \boldsymbol{H}_{m_D}^{(D-1)} & \boldsymbol{H}_{m_D+1}^{(D-1)} & \cdots & \boldsymbol{H}_{2m_D-1}^{(D-1)} \end{bmatrix}$$ |  | $\gamma \in \mathbb{R}^{K_1 \times \ldots \times K_D}$ |
| D-LEVEL BLOCK TOEPLITZ | $$\boldsymbol{T}^{(D)} = \begin{bmatrix} \boldsymbol{T}_{m_D}^{(D-1)} & \cdots & \boldsymbol{T}_2^{(D-1)} & \boldsymbol{T}_1^{(D-1)} \\ \boldsymbol{T}_{m_D+1}^{(D-1)} & \cdots & \boldsymbol{T}_2^{(D-1)} & \boldsymbol{T}_2^{(D-1)} \\ \vdots & \ddots & \vdots & \vdots \\ \boldsymbol{T}_{2m_D-1}^{(D-1)} & \cdots & \boldsymbol{T}_{m_D+1}^{(D-1)} & \boldsymbol{T}_{m_D}^{(D-1)} \end{bmatrix}$$ |  | $\gamma \in \mathbb{R}^{K_1 \times \ldots \times K_D}$ |
| D-LEVEL BLOCK HANKEL–TOEPLITZ, LEVEL D IS HANKEL | $$\boldsymbol{G}^{(D)} = \begin{bmatrix} \boldsymbol{G}_1^{(D-1)} & \boldsymbol{G}_2^{(D-1)} & \cdots & \boldsymbol{G}_{m_D}^{(D-1)} \\ \boldsymbol{G}_2^{(D-1)} & \boldsymbol{G}_3^{(D-1)} & \cdots & \boldsymbol{G}_{m_D+1}^{(D-1)} \\ \vdots & \vdots & \ddots & \vdots \\ \boldsymbol{G}_{m_D}^{(D-1)} & \boldsymbol{G}_{m_D+1}^{(D-1)} & \cdots & \boldsymbol{G}_{2m_D-1}^{(D-1)} \end{bmatrix}$$ |  | $\gamma \in \mathbb{R}^{K_1 \times \ldots \times K_D}$ |
| D-LEVEL BLOCK HANKEL–TOEPLITZ, LEVEL D IS TOEPLITZ | $$\boldsymbol{G}^{(D)} = \begin{bmatrix} \boldsymbol{G}_{m_D}^{(D-1)} & \cdots & \boldsymbol{G}_2^{(D-1)} & \boldsymbol{G}_1^{(D-1)} \\ \boldsymbol{G}_{m_D+1}^{(D-1)} & \cdots & \boldsymbol{G}_3^{(D-1)} & \boldsymbol{G}_2^{(D-1)} \\ \vdots & \ddots & \vdots & \vdots \\ \boldsymbol{G}_{2m_D-1}^{(D-1)} & \cdots & \boldsymbol{G}_{m_D+1}^{(D-1)} & \boldsymbol{G}_{m_D}^{(D-1)} \end{bmatrix}$$ |  | $\gamma \in \mathbb{R}^{K_1 \times \ldots \times K_D}$ |

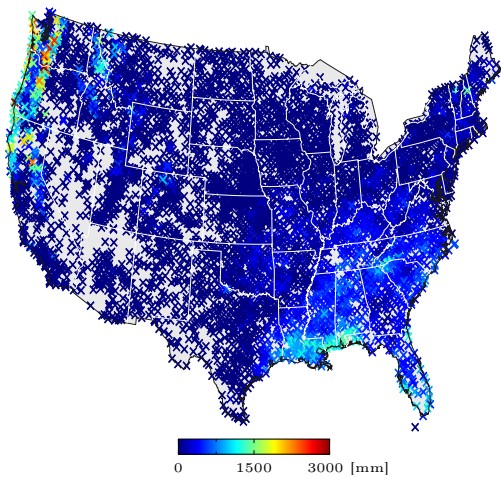

Figure A7: Raw data from precipitation data set. Each data point is visualized as a cross with color indicating the precipitation. The data set contains mostly low frequency content with some high frequency content apparent in the west coast as well as the south eastern parts.

## F.2 Magnetic Field Mapping

The magnetic field data has $N \approx 1.4$ million data points and is collected with an underwater vehicle outside the Norwegian coast. The data used were collected by MARMINE/NTNU research cruise funded by the Research Council of Norway (Norges Forskningsråd, NFR) Project No. 247626/O30 and associated industrial partners. Ocean Floor Geophysics provided magnetometer that was used for magnetic data acquisition and pre-processed the magnetic data. The data was later split into a training set and test set, at a roughly 50% split. The nature of the data split is visualized in Fig. A8b. However, in practice, we selected the width of the test squares and the training squares smaller than the one displayed in the illustration and they are merely that big for visualization purposes. The illustration displays squares that are 0.01 latitudinal degrees wide and 0.03 longitudinal degrees tall, corresponding to approximately 1 km in Cartesian coordinates in this area. The split we actually used was squares which were 0.001 latitudinal degrees wide and 0.003 longitudinal degrees tall, corresponding to approximately 100 m in both directions in Cartesian coordinates in that area. GP regression with a squared exponential kernel is able to extrapolate for approximately one lengthscale, but will not necessarily give a very informative prediction one or two lengthscales away from the nearest measurement. Although we do not know the lengthscale that would optimize the likelihood of the data before using training data to find it, we see from a zoomed-in version of Fig. 5b approximately how fast the magnetic field is changing across the spatial dimension and use this to make a reasonable guess at the distance we expect a well-tuned GP to be able to extrapolate the learned magnetic field. We then project the data into a local coordinate system using WGS84 and perform regression in Cartesian coordinates. We center and standardize the data with the mean and standard deviation of the training data. A squared-exponential kernel was used and we optimize the MLL in GPJax (Pinder & Dodd, 2022) using Adam (Kingma & Adam, 2015) for 100 iterations to find hyperparameters. The hyperparameters were initialized as $l = 200\,\mathrm{m}, \sigma_{\mathrm{SE}}^2 = 1$ and $\sigma_{\mathrm{y}}^2 = 1$. The resulting hyperparameters were $l_{SE} = 190$, $\sigma_{\mathrm{y}}^2 = 0.0675$, $\sigma_{\mathrm{SE}}^2 = 7.15$.

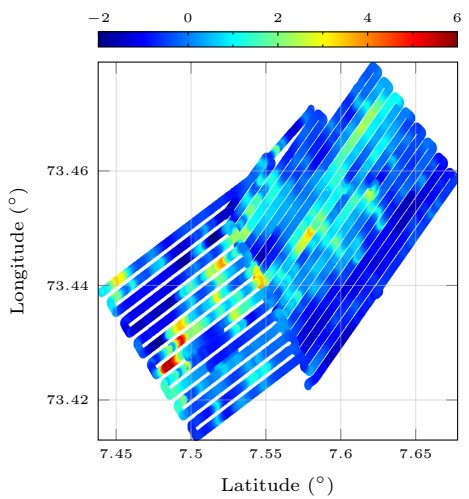

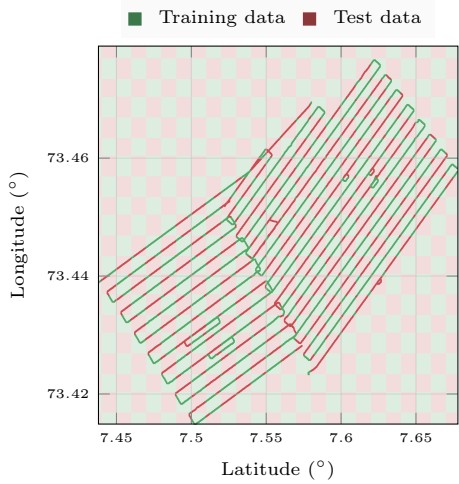

(a) Raw magnetic field measurements for the underwater magnetic field data. The plotted data is subsampled to $100^{\text{th}}$ of the data for visualization purposes.

(b) Data divided into training and test set. Deterministically split to ensure ensure that the lengthscale is captured properly in the training data.

Figure A8: Magnetic field training data and test data. The data is roughly 50/50 split between training and test set. Both training and test data are normalized only by the mean and standard deviation of the training data.

