# OpenReview forum: "Exploiting Hankel-Toeplitz Structures for Fast Computation of Kernel Precision Matrices"
_TMLR — Accepted by TMLR_

### Review · Reviewer_Kfc8 · 2024-05-27

**Summary Of Contributions:**

This paper provides a method to reduce the computational complexity of calculating the precision matrix in Hilbert-space Gaussian Processes without introducing approximations. For many kernels, the precision matrix is the sum of Hankel-Toeplitz matrices, that have $O(M)$ unique entries. Therefore, the computational complexity can be reduced from $O(NM^2)$ to $O(NM)$, where $N$ is the number of data points and $M$ is the number of basis functions. Experiments are done on synthetic and real datasets confirming their theoretical claims.

**Audience:**

Yes

**Broader Impact Concerns:**

The authors state in an impact statement that since their work proposes a new numerical method, it has many potential uses and there is no need to specifically highlight certain societal impacts. I agree; since the proposed method is general-purpose, it is difficult to assign either positive or negative impacts.

**Claims And Evidence:**

Yes

**Requested Changes:**

Critical:
- Improving the exposition as suggested in the previous question.

**Strengths And Weaknesses:**

### Strengths ###

- The paper is well-written. The mathematical reasoning is presented clearly.
- Experiments are done on real world datasets that confirm the theoretical claims.
- The proposed algorithm appears to be mathematically sound and applicable to a variety of kernels.

### Weaknesses ###
The exposition feels incomplete to me.
- The proposed method decreases the computational complexity of calculating the precision matrix. How does it affect the entire posterior mean/covariance computation (if any)?
- It would be helpful to readers to include more detailed pseudocode, if only for one type of kernel.

---

> ### Author Response · Authors · 2024-06-18
>
> We thank the reviewer for the evaluation of our work and the provided remarks. Below, we address each concern individually.
>
> >The proposed method decreases the computational complexity of calculating the precision matrix. How does it affect the entire posterior mean/covariance computation (if any)?
>
> Determining the posterior mean $\mu_{\star}(x_\star)$ or covariance $\Sigma_{\star}(x_\star,x_\star')$ when dealing with basis function approximations of the kernel consists in computing the following expressions:
> $$
> \begin{aligned}
> \mu_{\star}(x_\star) &= \phi(x_\star)^\top \left( \Phi^\top\Phi + \sigma^2\Lambda^{-1} \right)^{-1}\Phi^\top y,\\\\
> \Sigma_{\star}(x_\star,x_\star')&=\sigma^2 \phi(x_\star)^\top\left(\Phi^\top\Phi + \sigma^2\Lambda^{-1} \right)^{-1}{\phi}(x_\star').
> \end{aligned}
> $$
>
> Both expressions crucially require to compute the following quantity:
> $$
> (\Phi^\top \Phi + \sigma^2\Lambda^{-1})^{-1}.
> $$
>
> Computing the first term $\Phi^T \Phi$ (the precision matrix) naively requires $NM^2$ computations. Summing the the second term requires       $M^2$ computations. Finally computing the inverse of the sum requires $M^3$ computations. The total number of computations required by the naive approach is then $(N+1)M^2+M^3$. If we have more samples than basis functions (which is the required conditions for there to be computational savings), i.e., $N \gg M$, the computational complexity is of $O(NM^2)$. The inversion costs $O(M^3)$ are then negligible with respect to the cost of computing the precision matrix. Since our contribution reduces the cost of computing the precision matrix to $O(NM)$, it follows that the computational complexity of computing the posterior mean or covariance is also of $O(NM)$. We have clarified this point in the background on basis function approaches and also in the outlook and practical use (Section 3.5).
>
> >It would be helpful to readers to include more detailed pseudocode, if only for one type of kernel.
>
> The pseudocode in Algorithm 1 has been extended to accommodate this. The pseudocode now contains the computational complexity of all the steps discussed above. We remark that the particular kernel in use only affects what basis function should be used for approximation and doesn't change the general structure of the computations of Algorithm 1.
>
> >Improving the exposition as suggested in the previous question.
>
> We hope that our remedies for the reviewer's concerns has improved the exposition. We are more than happy to accommodate other suggestions as well.

---

### Review · Reviewer_2CuJ · 2024-05-31

**Summary Of Contributions:**

The submitted contribution focuses on the speed of the computation of kernel precision matrices by the exploitation of their possible Hankel-Toeplitz structure. In particular, the analysis is developed in the context of the Hilbert-space Gaussian Process approach, in which a covariance matrix is estimated from a dataset of $N$ data points by projecting a Gaussian process on a suitable set of $M$ basis functions. The standard technique requires $\mathcal O(NM^2)$ steps for the estimation of the precision matrix: as the needed number of basis functions might be large, the computational cost might be demanding. The authors show that a block Hankel-Toeplitz structure assumption on the precision matrix allows for a speed-up of the analysis, reducing the complexity to $\mathcal O(NM)$.

**Audience:**

Yes

**Broader Impact Concerns:**

There are no concerns regarding this publication, which is of theoretical/algorithmic nature.

**Claims And Evidence:**

Yes

**Requested Changes:**

- In the introduction, it is claimed that the paper ```We show that each increment of the HGP precision matrix can be split in a multilevel block-Hankel and a multilevel block-Toeplitz matrix```: I am not sure if this sentence is accurate. From what I understand, the specific Hankel-Toeplitz structure considered in the paper is not universal in the HGP approach. If this is the case, I would suggest to modify the sentence.
- In Eq. 5, is $\boldsymbol \phi^{(d)}(\boldsymbol x)$ to be replaced with $\boldsymbol \phi^{(d)}(x^{(d)})$?
- There are some notation imprecisions after Eq. 9 (e.g., $\boldsymbol H_k^{D-1}$ should be $\boldsymbol H_k^{(D-1)}$). Also, the notation in the labels in Fig. 3 does not look consistent with the text: please check.
- Section 3.3 seems to deal with block Hankel matrices only: if the authors meant to refer to block Hankel-Toeplitz matrices (as the context, the title, and the appearance of $\boldsymbol G^{(d)}$ suggest), they should specify it. This is actually quite a crucial point to clarify.
- In Eq. 19 and in Appendix B, the notation $\pi_{j_d}$ appears: do the authors mean $\pi j_d$? Please fix.
- In Eq. 20 the index $n$ in  the summand disappear.
- In Section 4, at the end of the first paragraph the authors refer to the computation of $2M$ and $3M$ components: were the experiments performed using $D=1$? (From Appendix F.1, it does not seem so). Please clarify.
- The writing of Section 5 might be improved (just to mention one aspect, the expression ```structure we exploit``` appears five times in two paragraphs...)
- In the main text *Corollaries* are introduced for specific bases, but in the Appendix, titles of corresponding proofs refer to the main Theorems instead of the corresponding Corollaries. Please fix.
- What does the subscript $t$ in Eq. 28 refer to? Why does the dependence on $d$ disappear from $L$ in the subsequent formulas?
- Appendix E is **empty**! If it was supposed to contain Table A1 (as it seems) please add one line to reference the content.

**Strengths And Weaknesses:**

*Strengths*
The speed-up presented by the paper can be beneficial in a number of settings that satisfy the hypothesis of the main theorems in the publication. The authors further presented some numerical experiments on real-world datasets, showing the benefit of their structured approach.

*Weaknesses*
The paper focuses on a specific, yet important, structure of the basis function approximation scheme.
Moreover, the writing of the paper can benefit from careful re-reading, as some parts are not, in my opinion, very clear.
In some aspects regarding the writing, the manuscript is unfortunately sloppy: I added below a list of changes, some of which I think are necessary for a publication.

---

> ### Author Response · Authors · 2024-06-18
>
> We thank the reviewer for their positive view, carefully reading our manuscript, and providing an exhaustive and precise list of detailed typos and mistakes in our manuscript. Below we provide a list of the changes implemented to fix these issues.
>
> >In the introduction, it is claimed that the paper "We show that each increment of the HGP precision matrix can be split in a multilevel block-Hankel and a multilevel block-Toeplitz matrix": I am not sure if this sentence is accurate. From what I understand, the specific Hankel-Toeplitz structure considered in the paper is not universal in the HGP approach. If this is the case, I would suggest to modify the sentence.
>
> The specific Hankel-Toeplitz structure considered in the paper is indeed not universal in the HGP approach. It only applies if the basis functions are placed on a cubical domain, rather than on any arbitrarily shaped domains. The sentence has been modified to reflect this fact.
>
> >In Eq. 5, is $\boldsymbol \phi^{(d)}(\boldsymbol x)$ to be replaced with $\boldsymbol \phi^{(d)}(x^{(d)})$?
>
> $\boldsymbol \phi^{(d)}(\boldsymbol x)$ has been changed to $\boldsymbol \phi^{(d)}(x^{(d)})$ in the uploaded revision.
>
> >There are some notation imprecisions after Eq. 9 (e.g., $\boldsymbol H_k^{D-1}$ should be $\boldsymbol H_k^{(D-1)}$). Also, the notation in the labels in Fig. 3 does not look consistent with the text: please check.
>
> $\boldsymbol H_k^{D-1}$ has been changed to $\boldsymbol H_k^{(D-1)}$, and $\boldsymbol H_k^{1}$ has been changed to $\boldsymbol{H}_{k}^{(1)}$. Also, the notation in the labels in Fig. 3 has been changed from $\boldsymbol H_d$ and $\boldsymbol T_d$ to $\boldsymbol H^{(1)}$ and $\boldsymbol T^{(1)}$
>
> >Section 3.3 seems to deal with block Hankel matrices only: if the authors meant to refer to block Hankel-Toeplitz matrices (as the context, the title, and the appearance of $\boldsymbol G^{(d)}$ suggest), they should specify it. This is actually quite a crucial point to clarify.
>
> In the original submission of the paper, Section 3.3 indeed dealt only with block-Hankel matrices. We apologize for this sloppy mistake. We have rewritten the section to demonstrate that the discussed property holds for block Hankel-Toeplitz matrices in general in the revised version.
>
> >In Eq. 19 and in Appendix B, the notation $\pi_{j_d}$ appears: do the authors mean $\pi j_d$? Please fix.
>
> We did indeed mean $\pi j_d$. This has been fixed in the revised version.
>
> >In Eq. 20 the index $n$ in the summand disappear.
>
> This problem has been fixed by replacing $\boldsymbol G^{(d),(1)}$ with $\boldsymbol G^{(d),(1)}(x_n^{(d)})$ as was done in Eq. 17, specifying that the term indeed changes with the index $n$.
>
> >In Section 4, at the end of the first paragraph the authors refer to the computation of $2M$ and $3M$ components: were the experiments performed using $D=1$? (From Appendix F.1, it does not seem so). Please clarify
>
> The experiments were indeed not performed using $D=1$. We defined $M$ to be the total number of basis functions (in the introduction). We defined $m_d$ to be the number of basis functions used to approximate each of the D components of the kernel (in the methods section, right before Eq. 4). In Eq. 6(a), the total number of basis functions $M$ corresponds to the number of entries in the vector $\boldsymbol\phi(x)$, while the number of basis functions $m_d$ is the number of entries in each component $\boldsymbol\phi^{(d)}(x^{(d)})$. From Eq. (6a), it follows that $M=\prod_{d=1}^{D} m_d$. Theorem 3.1 and 3.4 respectively require $\prod_{d=1}^D2m_d$ and $\prod_{d=1}^D3m_d$ components. Therefore, the experimental section should read $2^DM$ and $3^DM$, which has now been remedied.
> <!-- Therefore, the resulting computational complexities of $\mathcal{O}(\prod_{d=1}^{D} 2m_d)$ and $\mathcal{O}(\prod_{d=1}^{D} 3m_d)$ from the methods section become $\mathcal{O}(2^DM)$ and $\mathcal{O}(3^DM)$ respectively when expressed in terms of $M$. We have addressed this inconsistency in the experimental section. -->
>
> >The writing of Section 5 might be improved (just to mention one aspect, the expression structure we exploit appears five times in two paragraphs...)
>
> The related work section has been rewritten and the excessively repetitive phrasing has been removed.
>
> >In the main text Corollaries are introduced for specific bases, but in the Appendix, titles of corresponding proofs refer to the main Theorems instead of the corresponding Corollaries. Please fix.
>
> The appendix titles have been fixed so they now reference the corresponding corollaries rather than the theorems.

---

> ### Author Response · Authors · 2024-06-18
>
> >What does the subscript $t$ in Eq. 28 refer to? Why does the dependence on $d$ disappear from $L$ in the subsequent formulas?
>
> The subscript $t$ in (28) was a typo that has now been removed. The dependence on $d$ should remain and have been added to the subsequent formulas.
>
> >Appendix E is empty! If it was supposed to contain Table A1 (as it seems) please add one line to reference the content.
>
> Appendix E was indeed supposed to contain Table A1. A line has been added that references the content.

---

### Review · Reviewer_TQev · 2024-06-10

**Summary Of Contributions:**

The authors propose a way to speed up Gaussian Process inference when the precision matrix exhibits a special structure. Specifically they show that if the precision matrix can be decomposed in a combination of Hankel-Toeplitz matrices then they need to compute a number of parameters that scales linearly rather than quadratically with the number of basis functions used for the approximation of the precision matrix. This results in practice in speed ups of one or more orders of magnitude in the computation of the precision matrix, which they confirm in empirical studies.

**Audience:**

Yes

**Claims And Evidence:**

Yes

**Requested Changes:**

- a list of the use cases and applications that are most commonly addressed with the type of kernels that can be improved and of the cases that need to be modelled with kernels that are not subject to the speed up improvements
- a discussion on how these results could be generalised, i.e. how one could approach the problem of approximately (but automatically) decomposing a matrix in a composition of Hankel-Toeplitz matrices
- a comparison on the relative quality vs speed-up between the proposed approach and alternative approaches (e.g. using inducing point approximations)

**Strengths And Weaknesses:**

The results shown are of interest as the scheme proposed does not introduce additional approximations but is simply a more efficient way to perform the computation. The main contribution is in showing the sufficient conditions for the choice of the basis functions to exploit the prosed speed-up.

The weaker element of the proposed work are:
1. a clear illustration of the wider applicability and limitations of the results:
- the approach works for polynomial, complex exponential and sinusoidal basis functions with frequencies on a grid, for Hilbert–space Gaussian Process defined on a rectangular domain
- while the efficiency of other quadrature-like methods that are not defined on regular grids cannot be improved, e.g. approaches based on random frequency sampling or based on inducing points
- it would be useful for the reader to present a list of the use cases and applications that are most commonly addressed with the type of kernels that can be improved and of the cases that need to be modelled with kernels that are not subject to the speed up improvements

2.  It is not immediately clear how and if these results can be generalised, i.e. how one could approach the problem of approximately (but automatically) decomposing a matrix in a composition of Hankel-Toeplitz matrices

3. It would be useful to empirically show how the reduction of the computational and memory costs can allow the solution of problems of practical interest that were previously unfeasible (on a given computational budget), e.g. showing that a class of climate/weather predictions is now achievable on common hardware in reasonable timeframes.

---

> ### Author Response · Authors · 2024-06-18
>
> We thank the reviewer for the evaluation of our work and the provided remarks. Below, we address each concern individually.
>
> >a clear illustration of the wider applicability and limitations of the results
> > * the approach works for polynomial, complex exponential and sinusoidal basis functions with frequencies on a grid, for Hilbert–space Gaussian Process defined on a rectangular domain
> > * while the efficiency of other quadrature-like methods that are not defined on regular grids cannot be improved, e.g. approaches based on random frequency sampling or based on inducing points
> > * it would be useful for the reader to present a list of the use cases and applications that are most commonly addressed with the type of kernels that can be improved and of the cases that need to be modelled with kernels that are not subject to the speed up improvements
>
> Thank you for clearly summarising the wider applicability and limitations of our results. It is indeed the case that our results apply to polynomial, complex exponential, sinusoidal and Hilbert space Gaussian processes defined on a rectangular (hyper-cube) domain. Both stationary and non-stationary kernels can be approximated using said basis functions. Our approach indeed does not apply to inducing point or random frequency sampling methods. We have clarified this in the introduction and conclusion section of the paper.
>
> Regarding the nature of the kernel, its choice does not determine whether our algorithm can be used but rather the nature of the basis functions which are used for its approximation. Our approach thus does not rely on any particular kernel, but only focuses on the basis functions used to approximate said kernel. These considerations were stated in the results section and we have now repeated and refined them in the introduction to make it easier for a reader to understand the nature of our contribution.
>
> >It is not immediately clear how and if these results can be generalised, i.e. how one could approach the problem of approximately (but automatically) decomposing a matrix in a composition of Hankel-Toeplitz matrices
>
> Indeed, such an approximate decomposition would be extremely useful. However, it is not a trivial extension at all and requires careful consideration. As the idea itself is very interesting, we have added a discussion of how these results can hypothetically be generalized by automatic and approximate decomposition into Hankel-Toeplitz matrices to the future work section.
>
> >It would be useful to empirically show how the reduction of the computational and memory costs can allow the solution of problems of practical interest that were previously unfeasible (on a given computational budget), e.g. showing that a class of climate/weather predictions is now achievable on common hardware in reasonable timeframes.
>
> We agree that this would be of interest to the community. We believe that our precipitation experiment somewhat already addresses this concern. The original HGP paper already showed how that data set can now be processed in mere minutes on a common laptop CPU, which was certainly not the case at the time. Our paper takes this even further as can be seen in Figure 6(c\). Extrapolating these results to larger domains (which is the only difference for larger such experiments) should be immediate from this Figure.
>
> <!-- >a list of the use cases and applications that are most commonly addressed with the type of kernels that can be improved and of the cases that need to be modelled with kernels that are not subject to the speed up improvements -->
>
> >a comparison on the relative quality vs speed-up between the proposed approach and alternative approaches (e.g. using inducing point approximations)
>
> The proposed approach does not cause any degradation of quality, because the proposed approach contains no approximations to the HGP. It has the exact quality of any previously proposed approximation that our reformulation is compatible with. The HGP paper provides a comparison of the quality of the HGP approximation with the quality of several inducing point approximations. The original HGP paper further shows the computational benefits as well.
>
> The accuracy of our approach is proven in our paper to be exactly the same, because it gives the HGP prediction exactly. The reduction in computation time compared to HGP is investigated experimentally in the manuscript and can therefore easily be compared to other approaches through the original HGP paper.

---

### Decision · Action_Editor_LHjC · 2024-07-06

**Recommendation:** Accept as is

**Comment:**

This manuscript considers the problem of Gaussian process (GP) inference. A naive implementation of GP inference requires computational effort that grows cubically with the number of training observations, which limits scaling without further effort. This manuscript considers one approach to speed up the inferential process, namely the Hilbert-space Gaussian process (HGP), which operates via projection onto a set of basis functions. The authors show how to exploit convenient structure in HGP precision matrices to accelerate the computation required for the HGP approximation -- without any further approximation -- from scaling quadratically to linearly in the number of basis functions.

The reviewers were universally positive in their initial assessment of this work, although the initial reviews did identify a number of minor issues. These issues were adequately addressed during the author-reviewer discussion period, and the manuscript was strengthened as a result. Ultimately the reviewers agreed that the paper satisfies both TMLR acceptance criteria and were unified in their recommendation of acceptance. I agree with their assessment.

I encourage the authors to reflect on the reviewer feedback and author-reviewer discussion in preparing the camera-ready version of their manuscript.

**Audience:**

Yes, the central topic of this paper -- Gaussian process inference -- is a workhorse of Bayesian machine learning, and there is no question that a significant number of individuals in the TMLR audience would be interested in the findings of this paper.

**Claims And Evidence:**

Yes, following the author-reviewer discussion period, all reviewers agree that the claims in this manuscript are supported by accurate, convincing, and clear evidence.